# Frequency of Polymorphisms in *SLC47A1* (rs2252281 and rs2289669) and *SLC47A2* (rs34834489 and rs12943590) and the Influence of *SLC22A1* (rs72552763 and rs622342) on HbA1c Levels in Mexican-Mestizo Patients with DMT2 Treated with Metformin Monotherapy

**DOI:** 10.3390/ijms26178652

**Published:** 2025-09-05

**Authors:** Milton Abraham Gómez-Hernández, Adiel Ortega-Ayala, Oscar Rodríguez-Lima, Abraham Landa, Gustavo Acosta-Altamirano, Juan A. Molina-Guarneros

**Affiliations:** 1Programa de Doctorado en Ciencias Biomédicas, National Autonomous University of Mexico, Mexico City 04510, Mexico; abrahamgh70@comunidad.unam.mx; 2Department of Pharmacology, Faculty of Medicine, National Autonomous University of Mexico, Mexico City 04510, Mexico; ad.ortega@unam.mx; 3Department of Microbiology and Parasitology, Faculty of Medicine, National Autonomous University of Mexico, Mexico City 04510, Mexico; orodriguez@facmed.unam.mx (O.R.-L.); landap@unam.mx (A.L.); 4Dirección de Investigación, Hospital General de México “Dr. Eduardo Liceaga”, Mexico City 06720, Mexico; gustavo.acostaa@salud.gob.mx

**Keywords:** diabetes type 2, metformin, MATE1, MATE2, OCT1, polymorphisms, glycaemic control, HbA1c

## Abstract

Diabetes type 2 (DT2) entails significant health, economic, and productivity repercussions around the world. Poor glycaemic control, defined as an HbA1c >7.0%, has been associated with a number of complications. In spite of the large share of healthcare resources allocated to DT2 treatment, the proportion of controlled Mexican patients is among the lowest in the world (34.4%). Certain protein-encoding genetic polymorphisms involved in metformin transport may affect glycaemic control. We focused on determining the frequency of rs2289669, rs2252281, rs12943590, and rs34834489 polymorphisms in Mexican-Mestizo patients from the Tertiary Care Regional Hospital of Ixtapaluca, State of Mexico, Mexico, as well as assessing their possible association with therapeutic efficacy, as estimated through glycated haemoglobin. The individual polymorphism analysis did not reveal an association with glycaemic control; however, when combined with rs72552763 and rs622342, we found a significant positive correlation between HbA1c levels and metformin dose, which prevailed among patients carrying allelic variants of rs2289669 or rs12943590 who were also simultaneously carrying allelic variants of rs72552763 or rs622342. Patients carrying the reference allele of rs34834489 reported a significant positive correlation between HbA1c levels and metformin dose as well, regardless of their rs72552763 or rs622342 genotype. Thus, we identified alleles and allelic combinations of *SLC47A1*, *SLC47A2*, and *SLC22A1* polymorphisms posing a potential glycaemic control risk in Mexican-Mestizo patients.

## 1. Introduction

Diabetes refers to a group of metabolic diseases characterised by elevated blood glucose concentrations, known as hyperglycaemia. They are caused by either deficient insulin production or an organic lack of response to this hormone [1,2]. In 2021 these diseases were ranked as the eighth leading cause of death in the world, and by 2024 they were affecting 588.7 million people, entailing a treatment expenditure of USD 1.02 billion [3,4]. They also represent a loss of 2.3 healthy days per individual estimated over the world’s population [5].

The prevailing variant is diabetes type 2 (DT2), which accounts for up to 90% of the total number of cases [1]. DT2 causes a gradual increase in hyperglycaemia, which typically develops without symptoms until it manifests in an advanced stage characterised by polyuria, polydipsia, and idiopathic weight loss [1,2]. Once these symptoms are evident, a DT2 diagnosis is based upon either HbA1c or fasting glucose levels, registered as part of a random sample or at the end of a glucose-tolerance test [2,3].

The gold standard for monitoring the disease’s evolution is glycated haemoglobin (HbA1c) determination [6,7]. This biomarker reflects a patient’s average glycaemia over the course of the last 2–3 months [1]. An increase in HbA1c levels has been shown to be associated with a higher risk of developing various diabetes-related complications, such as nephropathy, retinopathy, neuropathy, and death from cardiovascular disease [8,9,10]. Therefore, glycaemic control and monitoring are key to maintaining the patient’s quality of life, where an HbA1c <7.0% is the goal [1,7,11]. Despite this, the proportion of controlled patients is highly variable and particularly low among Latin American populations [12]. In Mexico, only 34.4% of treatment-abiding patients who regularly attend medical consultations are under glycaemic control [13].

Glycaemic control can be influenced by various factors, including pharmacological treatment and its duration [14]. The first-line treatment against DT2 is metformin, a base débil which, being protonated at any physiological pH (1.3–7.8), is transported across body compartments via facilitated diffusion [15]. Organic cation transporters 1 and 3 (OCT1 and OCT3) enable the entry of metformin into hepatocytes, where it inhibits gluconeogenesis without undergoing metabolic modifications [16,17]. Meanwhile, organic cation transporter 2 (OCT2) and the multidrug and toxin extrusion proteins 1 and 2 (MATE1 and MATE2) are responsible for the renal elimination of metformin molecules [16].

Given the role of OCT1, OCT2, OCT3, MATE1, and MATE2 in the pharmacokinetics of metformin, their genetic variants, *SLC22A1*, *SLC22A2*, *SLC22A3*, *SLC47A1*, and *SLC47A2*, have been shown to be associated with changes in metformin therapeutic response. The GAT allele of the rs72552763 polymorphism in the *SLC22A1* gene has been associated with poor glycaemic control, as it appears more frequently in patients reporting this outcome [18]. A poor treatment response has also been associated with the A allele of rs622342 in *SLC22A1*, as diabetic patients from Chihuahua, Mexico, carrying this allele showed higher HbA1c levels compared to C homozygotes [19]. However, this differs from other reports which suggest that the ancestral A allele of rs622342 is associated with a favourable response to metformin treatment [20,21].

To some extent, this contrast between studies reflects the modification of certain alleles’ effects as induced by the simultaneous presence of other polymorphisms’ alleles. Thousands of polymorphisms have been reported in *SLC47A1* and *SLC47A2* [22]; however, in most cases, the variant allele is present in less than 25.0% of the global population, and among those present in more than 25.0%, only a few have been associated with changes in glycaemic control in patients with DMT2. In this context, the variant alleles of polymorphisms rs2289669, rs2252281, rs12943590, and rs34834489 are present in at least 26% of the global population and have been shown to be related to glycaemic control in patients with DMT2 [22,23,24,25,26,27]. The A allele of rs2289669 in *SLC47A1* has been associated with glycaemic control, as it reduces HbA1c levels by 0.30% for each A carried [25]. This effect increases when allele A of rs2289669 in *SLC47A1* is present along CC of rs622342 in *SLC22A1* [26]. A similar case is observed with C homozygotes of rs2252281, when patients carry the reference allele GAT of rs72552763 in *SLC22A1* [27].

The simultaneous presence of the MATE1 and MATE2 genotypes has also been associated with changes in pharmacokinetic parameters. The reference allele T homozygotes of rs2252281 in *SLC47A1* showed an increased metformin renal clearance when A of rs12943590 in *SLC47A2* was simultaneously present as compared to G homozygotes of rs12943590 [27]. Renal clearance has also been found to increase in patients carrying both A of rs12943590 and A of rs34834489—a variant in *SLC47A2* which also contributes independently to increased renal clearance [24].

In this context, the aim of the present study was to determine whether the genotypes of *SLC47A1* (rs2252281 and rs2289669) and *SLC47A2* (rs12943590 and rs34834489) are associated with the therapeutic response to metformin when they are substratified according to different genotypes of rs72552763 and rs622342 in *SLC22A1* of OCT1. Accordingly, our research team genotyped the polymorphic variants of *SLC47A1* and *SLC47A2* within a sample of DT2 Mexican-Mestizo patients (DT2MM) from the Regional High Speciality Hospital of Ixtapaluca (HRAEI) who had previously been genotyped for rs72552763 and rs622342. Analyses simultaneously considering *SLC47A1*, *SLC47A2*, and *SLC22A1* genotypes report a shift in the positive correlation between HbA1c levels and defined daily dose (DDD) in mg/kg/day.

## 2. Results

### 2.1. SCL47A1 and SLC47A2 Allelic and Genotypic Frequencies

Initially, without therapy stratification (metformin monotherapy or metformin + glibenclamide), 204 DT2 patients were genotyped for rs2252281, rs2289669, rs12943590, and rs34834489. One patient was not genotyped for rs34834489 due to an absent sample. Table 1 shows the genotypes of all polymorphisms in Hardy–Weinberg equilibrium.

The comparison of the genotypic frequencies of rs2252281, rs2289669, rs12943590, and rs34834489 in DT2MM patients and those reported in the 1000 Genomes Project shows differences with respect to the world’s population in general and some specific populations. As shown in Figure 1, the genotypic frequencies of the four studied polymorphisms in our HRAEI sample are statistically different (*p* < 0.001) from those observed in the African population. Likewise, the genotypic frequencies of rs2252281, rs2289669, and rs12943590 in our sample are statistically different (*p* < 0.05) from those reported specifically among Europeans.

Unusually, genotypic frequencies of rs2252281 and rs12943590 reported in American populations are statistically different (*p* < 0.05) from those observed in our sample. Meanwhile, the HRAEI DT2 patients presented similar genotypic frequencies to those reported in Asian populations, where all four variants coincide with South Asians and rs2289669, rs12943590, and rs34834489 coincide with Eastern Asians.

### 2.2. Clinical and Demographic Characteristics Across Different Genotype Carriers Undergoing Metformin Monotherapy

The analysis of *SLC47A1* and *SLC47A2* polymorphisms was restricted to patients undergoing metformin monotherapy. Out of the 204 patients sampled, we identified 59 individuals receiving this treatment. Their general clinical and demographic characteristics had been previously described [23].

These metformin monotherapy patients were stratified according to their genotypes for rs2252281, rs2289669, rs12943590, and rs34834489. As shown in Appendix A, no significant differences were found in HbA1c levels, glycaemic control, or other clinical and demographic variables between the genotypes of the same polymorphism, except for height, where differences were observed for the genotype of rs34834489. It should be noted that, after applying the Bonferroni correction for multiple comparisons, no significant differences were found. across the different genotypes of rs2289669 (Appendix A), rs12943590 (Appendix A) and rs34834489 (Appendix A).

### 2.3. Correlation Between HbA1c Levels and Daily Dosage

The gradual increase in metformin monotherapy dosage aims to lower HbA1c to target glycaemic control levels. Under this assumption, we verified whether there was a correlation between HbA1c levels and metformin daily dose (DDD, mg/kg/day). Figure 2a shows that HbA1c levels and metformin daily dosage (DDD, mg/kg/day) are positively correlated (*p* < 0.05). Since dose increases result in higher metformin plasmatic concentrations [28], we evaluated the correlation between these two variables; however, as shown in Figure 2b, no significant correlation was observed between them.

Subsequently, we examined whether the positive correlation between HbA1c levels and DDD (expressed as metformin mg/kg/day) remained significant regardless of genotype for rs2252281, rs2289669, rs12943590, and rs34834489. Appendix A shows that the correlation between HbA1c and metformin dose is positive and remains significant in rs2289669 G allele homozygotes, rs12943590 heterozygotes, G allele homozygotes, and rs34834489 heterozygotes. The correlation did not remain significant in rs2252281, as shown in Appendix A.

Considering the possibility of dominant alleles, a second analysis was carried out using the dominant genotype approach. A dominant allele was defined as one present in linear correlation lines with similar slopes or directions. Figure 3 shows that in dominant allele carriers of rs2289669, rs12943590, and rs34834489 the correlation between HbA1c levels and metformin DDD remains significant. Additionally, Figure 3a shows that this correlation is wider in homozygotes of the rs2289669 G allele, although the correlation was not significant. In contrast, in Figure 3b,c, the correlation is negative in homozygotes of the rs1294359 G allele and homozygotes of the rs34834489 A allele, but it is not significant.

### 2.4. Correlation Changes Between HbA1c Levels and Daily Dose Induced by Simultaneous Genotypes

Since previous studies have proven that the presence of simultaneous genotypes of different polymorphisms may change individual responses [18,26,27], we decided to explore this approach.

Our research team had previously identified that the alleles of rs622342 and rs72552763 are associated with increased survival under inadequate glycaemic control [20]. Therefore, we revisited the dominant alleles of rs2289669, rs12943590, and rs34834489, which sustained a significant correlation between HbA1c levels and the metformin DDD (Figure 3), and we then stratified patients according to the dominant and recessive alleles for rs72552763 and rs622342 (Figure 4 and Figure 5).

As shown in Figure 4 and Figure 5a–c, among carriers of the A allele of rs2289669 or rs12943590, the correlation between HbA1c levels and metformin DDD is positive and remains significant only when patients are also carrying the dominant allele of rs72552763 or rs622342. When patients are homozygous for the GAT or A alleles of rs72552763 or rs622342, the correlation between the variables becomes negative, yet not significant. Finally, as can be seen in Figure 5d–f, among carriers of the G allele of rs34834489, the correlation between HbA1c levels and metformin daily dose is positive and remains significant both among dominant allele carriers of rs72552763 or rs622342, as well as in the GAT and A homozygotes of each respective polymorphism.

## 3. Discussion

### 3.1. Genotypic Frequency Comparison Against Other World Populations

The 1000 Genomes Project is a reference study in population genetics research conducted on African, Eastern Asian, Southern Asian, European, and American populations. Its main objective is to elucidate the properties and distribution of both common and rare genetic variants [29]. We identified differences between the genotypic frequencies of rs2252281, rs2289669, rs12943590, and rs34834489 observed in HRAEI DT2MM patients and those reported in the 1000 Genomes Project for other populations. The difference in genotypic frequencies of rs2252281, rs2289669, and rs12943590 within our sample and other global populations may be explained by the genotypes’ over-representation, inasmuch as the genotypic frequencies reported by Ensembl for the world’s population are data for SAS, EAS, AMR, EUR, and AFR ancestries [22]. In our sample, DT2MM patients presented frequencies of rs2252281, rs2289669, and rs12943590 which differed from those of at least two of the aforementioned ancestries.

Moreover, the frequency difference observed in our sample with respect to other populations may be the result of the under-representation of DT2MM individuals in the 1000 Genomes project. In that project Los Angeles Mexicans are incorporated into the wider Mexican population [29] without accounting for the environmental differences the aforementioned group is exposed which, in turn, might present a genetic divergence from Mexican-Mestizos. This would also explain the frequency differences between rs2252281 and rs12943590 reported among DT2MM and AMR, given that the 1000 Genomes project integrated the latter group with Los Angeles Mexicans, Colombians, Peruvians, and Puerto Ricans [29].

On the other hand, the differences between the genotypic frequencies of the four studied polymorphisms in DT2MM patients and the African population are supported by ancestry studies in healthy individuals and patients with diabetes, where African ancestry ranges between 5.0 and 12.0% [30,31]. These studies might also explain the differences observed between DT2MM and EUR populations regarding rs2252281, rs2289669 and rs12943590 in view of the fact that European ancestry in Mexico City exhibits very wide variation (0.0–55.0%). Nevertheless, the similarity between the genotypic frequencies in our sample and in the Asian population contrasts with this ancestry’s low reported rate in the Mexican population [30]. This similarity could be explained by the difference in the number of polymorphisms we evaluated as compared to the evaluation by Sohail et al. conducted in 2023 [30]; our work did not propose an ancestry study.

These genotypic frequency differences observed for rs2252281, rs2289669, rs12943590, and rs34834489 in the DT2MM population could impact its representation in clinical studies of those molecules whose transport depends on MATE1 and MATE2, such as metformin. In fact, the Mexican population is not widely represented across clinical studies, as the records available at Trials.gov from 2000 to 2020 report Latino patient recruitment of only 6.0%, whilst 79.7% recruited Caucasians and 10.0% recruited Africans [32].

### 3.2. No Association Between Polymorphisms and Metformin Therapeutic Efficacy

Our study’s main objective was to determine the association between the therapeutic efficacy of metformin monotherapy and polymorphisms across the genes *SLC47A1* (rs2252281 and rs2289669) and *SLC47A2* (rs12943590 and rs34834489) among DT2MM patients. When we analysed these polymorphisms separately, we found no significant differences in HbA1c levels or glycaemic control across genotypes. This coincides with the absence of an association between rs2252281 and the relative change in HbA1c found by Stocker et al. in 2013 [27] among Caucasians and African-Americans. The absence of significance also matches the reported absence of a difference in glucose and HbA1c levels across rs2289669 and rs12943590 genotypes among DT2MM patients [18]. Neither of these two polymorphisms affected glycaemic control in the Southern Indian population, as the relative change difference in HbA1c levels was not significant across the polymorphisms’ genotypes [33]. The alternative selection of glucose as a biomarker in individual analyses has not yielded differences in glucose levels across the various genotypes of rs2252281, rs2289669, and rs12943590 [34]. This seems to explain why our initial analysis had a similar result. Individual analyses of glycaemic control associated with *SLC47A1* and *SLC47A2* polymorphisms has been reported, as only a few studies have associated them with glycaemic control under this approach. Chinese DT2 patients carrying the allelic variant of rs2289669 presented a significant decrease in HbA1c with respect to reference allele carriers [35]. An individual analysis of rs2289669 in *SLC47A1* and rs12943590 in *SLC47A2* among Chinese-Han DT2 patients showed an association with metformin response. Normal-weight patients carrying the homozygous reference allele and the variant of rs2289669 presented a greater decrease in fasting insulin and insulin resistance index (HOMA) as compared to heterozygotes [36]. In turn, a regression analysis performed by Chen et al. in 2022 [36] revealed a greater HOMA decrease among rs12943590 heterozygotes as compared to reference allele and variant homozygotes.

The lack of association between HbA1c levels and rs2252281 and rs2289669 in *SLC47A1* might be explained by their lack of effect on renal clearance, as observed in Danish individuals [37]. This could also be the case for rs12943590 and rs34834489 in *SLC47A2*, given that an individual analysis of healthy Korean volunteers reported no difference in renal clearance or maximum metformin concentration across these polymorphisms’ genotypes either [38]. However, these results, just like our own, contrast the renal clearance increment observed through haplotype analysis in Korean individuals carrying the homozygous reference alleles of rs758427 and rs34834489, as well as the homozygous allele of rs12943590 [24]. This approach could not be explored due to the absence of the rs758427 genotype as well as the reduced number of patients, which would have implied a low frequency of the aforementioned haplotypes.

### 3.3. Positive Correlation Between HbA1c Levels and Metformin Dose Among Patients Carrying Specific Polymorphisms

Assuming that a gradual increment in metformin dosage aims to reduce HbA1c to target levels of glycaemic control and that this increment entails a higher metformin plasmatic concentration [28,39], we assessed the correlation between these three markers. Thus, we identified a significant correlation between metformin dose and HbA1c levels. The lack of correlation between DDD and metformin plasmatic concentration could be due to the intrinsic response variance, which reflects metformin’s wide compartmentalisation in the human body [40]. A study on a sample of 467 patients showed that metformin plasmatic concentration presents a free distribution with a 1300 ng/mL median and an ample range from 0 ng/mL (untraceable) up to 113,000 ng/mL [41]. Given these considerations, the r^2^ of 0.019 obtained through the correlation analysis of DDD and metformin plasmatic concentration is altogether reasonable.

On the other hand, the significant positive correlation between DDD and HbA1c levels suggests that, in some patients, an increase in metformin dose does not correlate with a decrease in HbA1c. Genotype-stratified correlations of both variables remained significant across some genotypes of rs2289669, rs1294359, and rs34834489. The positive correlation between DDD andHbA1c levels in G homozygotes of rs2289669 in *SLC47A1* matches the lesser reduction in HbA1c levels observed among carriers of the rs2289669 reference allele [35].

The observed positive correlation across variables for rs129359 in *SLC47A2* contrasts with previous reports. A study on DT2 Chinese patients treated with metformin showed that rs1294359 heterozygotes present a greater HOMA-IR index reduction [36]. This HOMA-IR difference throughout genotypes cannot be directly ascribable to any particular allele, as A of rs1294359 has reportedly increased the activity of *SLC47A2’s* promoter in vitro and A homozygotes have shown an increased metformin clearance [24]; thus, heterozygotes would be expected to present an intermediate phenotype. Results obtained by Chung et al. match the preservation of the positive correlation between DDD and HbA1c levels found in A homozygotes of rs12943590 through the dominant genotypic model.

In turn, the observed positive correlation between DDD and HbA1c levels for G homozygotes and heterozygotes of rs34834489, while considering G as the dominant allele, diverges from the association of A as a non-control risk allele, which is due to its in vitro *SLC47A2* promoter increased activity as well as its increased metformin clearance among homozygotes [24]. The discrepancy might be due to Chung et al. analysing rs34834489 using a haplotype approach along rs758427 and rs1294359. This also suggests that rs12943590 might affect the impact of rs34834489, as neither of the haplotypes of the polymorphisms’ variants assessed by Chung et al. were simultaneously present.

Finally, the positive DDD-HbA1c correlations observed through a dominant genotypic model in both G homozygotes and allele A carriers of rs2289669 differ from previous studies, where A was reported as the protection allele [35,36]. When isolated, none of our results showed that rs2289669 alleles were correlated with DDD or HbA1c levels; however, as evidenced by the simultaneous polymorphism analysis, the observations from previous reports could be due to the presence of several polymorphisms across different transporter genes at the same time.

### 3.4. Simultaneous Presence of Polymorphisms and DDD-HbA1c Correlation Changes

Responses have differed when studied along other polymorphisms as opposed to individual analyses [18,25,27,37,42]. Since we had previously reported that allelic variant carriers present a lesser non-control survival median as compared to reference allele homozygotes of rs72552763 and rs622342 in *SLC22A1* [20], we decided to explore the aforementioned collective approach as we had the pertinent genotypes of all 59 patients. 

Among rs2289669 and rs12943590 allelic variant carriers who simultaneously carried either rs72552763 or rs622342 allelic variants, the correlation between HbA1c levels and metformin dose remained significant and positive; this correlation disappeared in rs72552763 and rs622342 reference allele carriers (*SLC22A1*). This suggests a possible influence of these polymorphisms on HbA1c levels. Our results match the increased reduction in HbA1c levels reported by rs2289669 allelic variant carriers (*SLC47A1*) who are also rs622342 allelic variant homozygotes (*SLC22A1*) [26].

Our results might explain the discrepancy between studies by Menjivar et al., 2020 [18], and Ortega-Ayala et al., 2024 [20]. Menjivar et al., 2020, reported a higher poor-control risk for rs72552763 and rs12943590, which were thus identified as risk alleles [18]. In turn, Ortega-Ayala et al., 2024, reported that variant alleles of rs72552763 and rs622342 present a lesser non-control survival median (677 and 718 days, respectively) [20]. The increment in *SLC47A2* translation [24,42] paired with non-functional *SLC22A1* alleles could lead to augmented metformin elimination in view of the effect combination. The lower hepatic volume of distribution observed in carriers of the DEL allele of rs72552763 [43] could increase the plasma concentration of metformin. When this is combined with an increase in metformin renal clearance, as observed in carriers of the variant alleles of the polymorphisms rs12943590 and rs34834489 [24,42], it could promote metformin elimination, thereby reducing its therapeutic efficacy, since metformin’s main action mechanism is a reduction in hepatic glucose production [17].

Our results evidence that not all of the polymorphisms we studied, either individually or in combination, exhibit any type of correlation between metformin dose and HbA1c. The response by the rs34834489 reference allele was unchanged by either reference alleles or variants of rs72552763 and rs622342. The reference allele of rs34834489 might entail non-control risk. Finally, even as metformin clearance changes have been reported for rs2252281 homozygotes simultaneously carrying the allelic variant of rs316019 in *SLC22A2* [37], we did not extend our analysis to rs316019 given that the allelic variant’s frequency was rather low among the 59 studied patients [23]. Our study presents several limitations that should be considered when interpreting its findings and to guide the design of future studies. The sample size (n = 204) was determined based on the reported global proportions of the polymorphisms, addressing the primary objective of determining the genotypic frequencies of polymorphisms rs2289669, rs2252281, rs12943590, and rs34834489, which was achieved. When stratifying the genotypes of the studied polymorphisms, some groups comprised fewer than 10 observations (GG of rs2289669 and AA of rs34834489); therefore, the correlation and regression analyses of HbA1c and DDD should be interpreted with caution. On the other hand, the sample size for correlations of the remaining genotypes met methodological recommendations [44,45]. Nevertheless, it was insufficient for performing multivariate analyses that could provide additional information.

Another limitation is that our study had a cross-sectional design, and we only had single time-point measurements of the biomarkers. Consequently, it was not possible to determine whether the findings remained consistent over time. Finally, the interval between the collection of peripheral venous blood samples and the evening metformin dose could vary between patients, potentially leading to wide variability in plasma metformin concentration.

## 4. Materials and Methods

### 4.1. Study Design and Patient Recruitment

A cross-sectional observational study was conducted on patients who had been previously diagnosed with DT2 according to American Diabetes Association (ADA) criteria [1]. A total of 204 DT2 Mexican-Mestizo patients over the age of 18 were included, all of whom were treated between April 2018 and April 2019 with pharmacological therapy for a minimum of three months. None reported kinship with one another, and everyone manifested at least three generations of Mexican ancestry. Each participant provided written informed consent. The study was conducted in accordance with the principles for medical research involving human subjects, as outlined in the Declaration of Helsinki and the 64th General Assembly of the World Medical Association (WMA), in October 2013 at Fortaleza, Brazil. Clinical and demographic characteristics at the time of enrolment were retrieved from medical records and electronic files. Each patient’s information was compiled in a database and verified through probabilistic random sampling.

### 4.2. Sample Delimitation

Out of the initial 204 candidates, 101 were excluded from the analyses for the following reasons: 2 plasmatic samples were insufficient, 14 were reported as unquantified, and 29 were reported as undetermined (total: 45), while a further 56 were lost during follow-up (41 changed treatment and 15 had no HbA1c record). Out of the remaining 103 included patients, plasmatic concentrations could only be determined in 86, since 9 had been registered as undetermined and 8 more were not quantified (either haemolysed or measured using lipemic serum). Out of these, 59 individuals undergoing metformin monotherapy and with available data on metformin dosage, HbA1C levels, and body weight, in addition to matching the previously outlined inclusion criteria, were selected. Patients with chronic alcoholism, previous pancreatic diseases, diabetes type 1 (DT1), renal failure, incomplete medical records, or those treated with insulin or insulin analogues were not included.

### 4.3. Genotyping

EDTA tubes were used to collect 10 mL of peripheral blood from each patient. Genomic DNA (gDNA) was then extracted from 200 μL of peripheral blood using the UltraClean^®^ BloodSpin^®^ DNA (MoBio Laboratories, Carlsbad, CA, USA) isolation kit, and the samples were stored at −20 °C. Prior to use, the gDNA was quantified using an EPOCH microplate spectrophotometer (Agilent Technologies, Santa Clara, CA, USA), and its integrity was verified through electrophoresis on a 2.0% agarose gel in tris-acetate-EDTA buffer solution. Genotypes of rs2252281, rs2289669, rs12943590, and rs34834489 were determined through real-time PCR using TaqMan-based fluorescence assays on a *StepOne Real-Time PCR* System (Applied Biosystems; Thermo Fisher Scientific, Inc., Waltham, MA, USA) platform. Reactions were performed on MircoAmp Fast Optical 48-Well Reaction Plates (Applied Biosystem; Thermo Fisher Scientific, Inc.) with a final reaction volume of 10 μL per well, using a 0.8X concentration of TaqMan Universal PCR MasterMix (Applied Biosystem; Thermo Fisher Scientific, Inc.), a 0.5X probe concentration (*SLC47A1*: rs2252281, C__15882235_10; *SLC47A1*: rs2289669, C__15882280_10; *SLC47A2*: rs12943590, C___2593951_10; *SLC47A2*: rs34834489, C___2593950_10), and 20 ng of gDNA as template. The amplification protocol consisted of a previous reading at 60 °C for 30 s and enzyme activation at 95 °C for 600 s, followed by 40 cycles of gDNA denaturation at 95 °C for 15 s and probe hybridisation at 60 °C for 90 s, with a final extension at 60 °C for 30 s. Independent genotyping assay data were analysed jointly using TaqMan Genotyper Software v 1.7 (https://www.thermofisher.com/mx/es/home/technical-resources/software-downloads/taqman-genotyper-software.html (accessed on 26 June 2024)), and integrated files containing the genotypes for each polymorphism were retrieved. Patient genotypes for rs72552763 and rs622342 in *SLC22A1* were taken from the previous work by Ortega-Ayala et al., 2022 [23].

### 4.4. Plasmatic Metformin Determination

Plasmatic concentrations could only be determined in 86 subjects, since 9 had been registered as undetermined and 8 more were not quantified (either haemolysed or measured through lipemic serum). Determinations were carried out in the Clinical Pharmacology Unit of UNAM’s Faculty of Medicine. The methodology was validated in accordance with the Mexican Official Normativity NOM-177-SSA 1-2013 [46], which establishes tests and procedures to demonstrate a drug’s interchangeability, the mandatory requirements authorised third parties must observe, which research or healthcare institutions may perform biocompatibility tests, and the internal procedure of analytical methodology validation. The study also adhered to additional international requirements, whose acceptance parameters were established in the Standard Operating Procedure SOPUA-05-09, “Validation of analytical methodology on special and bioavailability and/or bioequivalence studies”. To analyse biological samples, we employed the method described in the analytical methodology index card FMA-018/B, which had been previously validated according to Mexican Official Normativity NOM-177-SSA1-2013. The analytical method was selective over the quantification of plasmatic metformin without the interference of either endogenous or exogenous compounds. The employed methodology proved to be selective, linear, precise, and exact over the assessed concentration range. For sample analysis, we employed UHPLC-MS/MS in MRM mode using an Agilent Technologies G6490A mass spectrometer (Agilent Technologies Inc., Santa Clara, CA, USA). In the preparation of calibration curves and controls for sample analysis, we employed the following reference substances: metformin hydrochloride (U.S.P. batch: R069H0, purity: 99.7%), and loratadine (U.S.P.: R052U0, purity: 99.8%). To quantify plasmatic metformin, we employed a mass/charge ratio for metformin of 130.1/71.0, and that for the internal standard, loratadine, was 383.1/337.1. A Luna PFP analytical column (2.0 × 100 mm, 3.0 m) obtained from Phenomenex (Phenomenex S.r.l., Torrance, CA, USA) was used for analyte separation and determination. The isocratic elution of samples was carried out using acidified ammonium formate 10 mM (A)/acetonitrile 100% (B) as a mobile phase. Analytes were previously extracted through protein extraction/precipitation: A 100 μL aliquot was extracted from a plasma sample and subsequently deposited into a microtube. We added a 10 μL aliquot from the loratadine internal standard solution (30 g/mL). To carry out protein precipitation, we added a 400 μL aliquot of HPLC-grade acetonitrile. The tube was shaken on a multiple vortex at maximum speed for 1 min. The tube was centrifuged at 20,600× *g* and 4 °C for 5 min. We recovered 250 μL of the supernatant and transferred it to a 96-well plate. The injection volume on the chromatographic system was 2.0 μL. The method was linear in the range of 20–10,000 ng/mL. Intra-day and inter-day variation coefficients were less than 15%. Metformin recovery ranged from 89.676 to 90.731%. The relationship between chromatographic response and concentration on every calibration curve was adjusted through linear least-squares regression for metformin. To quantify the plasmatic samples, the regression was performed using Mass Hunter V: B.08 Quantitative Analysis software. Patients were summoned by their respective treating physicians, having observed a fasting period of at least 8 h. All of the blood samples were taken within an interval of 8 h after the evening’s metformin dose. A 10 mL peripheral venous blood sample was extracted using EDTA tubes. The sample was centrifuged at 400× *g* for 5 min at 4 °C. Once the plasma was obtained, aliquots were carried out using Eppendorf tubes, and the samples were frozen at −80 °C until drug determination assays were simultaneously performed across all of them.

### 4.5. Statistical Analysis

#### 4.5.1. Analysis of Allelic and Genotypic Frequencies

Based on integrated genotype files, VIC and FAM probe results were recodified in accordance with the respective alleles, and each polymorphism’s genotypic and allelic frequencies were determined through simple counts. Subsequently, language R v.4.2.2 (https://www.R-project.org/ (accessed on 4 November 2024)) was used to determine Hardy–Weinberg equilibria (HWEs) through chi-square tests, where the *p* value was based upon the cumulative distribution of the chi-square results with one degree of freedom; *p* > 0.05 was deemed indicative of HWE equilibrium. Genotypic frequencies for rs2252281, rs2289669, rs12943590, and rs34834489 from five different world populations included in the 1000 Genomes Project [29] were obtained from Ensemble [22] and compared against our DT2MM patient sample through a chi-square test.

#### 4.5.2. Initial Inferential Analysis

These analyses, like the rest, were carried out using language R, v.4.2.2. Initially, genotype-based patient subgroups were generated for rs2252281, rs2289669, rs12943590, and rs34834489. Variable distribution types within the subgroups were then evaluated using either Kolmogorov–Smirnov or Shapiro–Wilk tests. Depending on the distribution and the number of independent groups to be compared, quantitative variables were analysed using the Student’s *t*-test, the Mann–Whitney U test, ANOVA, or the Kruskal–Wallis test. Nominal variables were analysed using either chi-square or Fisher’s exact tests, depending on the number of observations and groups. The data were presented as means and standard deviations; 25th, 50th (median), and 75th percentiles; or frequencies, as needed.

#### 4.5.3. Linear Regressions

The linear correlation between HbA1c levels or metformin plasmatic concentration (response variable) and DDD (predictor variable) was assessed in the dataset, where this information was available. Subsequently, the linear correlation between HbA1c levels and DDD was evaluated throughout patients with different genotypes of rs2252281, rs2289669, rs12943590, and rs34834489. The linear correlation between HbA1c levels and DDD was also assessed over strata generated using a dominant genotype approach. Next, among individuals for whom a significant linear correlation between HbA1c levels and DDD was identified, a second stratification was performed based on carriers and non-carriers of the dominant alleles of rs72552763 and rs622342. Finally, the linear correlation between HbA1c levels and metformin daily dose expressed in mg/kg was evaluated over these substrata.

## 5. Conclusions

The genotypic frequencies of rs2252281, rs2289669, and rs12943590 found in DT2MM patients are different across the world’s population. The genotypic frequencies of rs2252281 and rs12943590 found in DT2MM patients are also different across the American population. *SLC47A1* and *SLC47A2* genotypes may result in an under-representation of DT2MM throughout clinical studies.

The individual analysis of rs2252281, rs2289669, rs12943590, and rs34834489 did not yield an association with therapeutic response. However, the collective analysis including the polymorphic variants of *SLC22A1*, rs72552763, and rs622342, conducted using a dominant genotypic model, allowed us to identify allelic combinations preventing poor glycaemic control risks. We observed no significant positive correlation between HbA1c levels and metformin dose among carriers of rs2289669 in *SLC47A1* and del rs12943590 allelic variants in *SLC47A2* who simultaneously carried the reference alleles of either rs72552763 or rs622342 in *SLC22A1*. Therefore, this particular allele combination might be relevant to therapeutic response as estimated by HbA1c levels. The rs34834489 reference allele in *SLC47A2* could imply a non-control risk, given that the correlation between HbA1c levels and metformin daily dose remained significant and positive when its carriers also carried either of the rs72552763 and rs62234 alleles in *SLC22A1*.

## Figures and Tables

**Figure 1 ijms-26-08652-f001:**
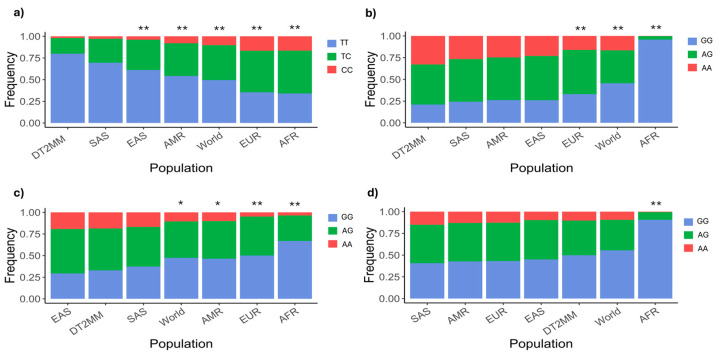
Genotypic frequencies of *SLC47A1* and *SLC47A2* polymorphisms. Comparison of the observed frequencies in the DT2MM sample and those reported in the 1000 Genomes Project for polymorphisms (**a**) rs2252281, (**b**) rs2289669, (**c**) rs12943590, and (**d**) rs34834489. Statistical differences relative to the DT2MM sample (*: *p* < 0.05, **: *p* < 0.001) were assessed using a chi-square test. The *p* value for each comparison was obtained through Monte Carlo simulation with 10,000 replicates. DT2MM: Mexican-Mestizo patients with diabetes type 2 from HRAEI, SAS: Southern Asians, EAS: Eastern Asians, AMR: Americans, EUR: Europeans, AFR: Africans.

**Figure 2 ijms-26-08652-f002:**
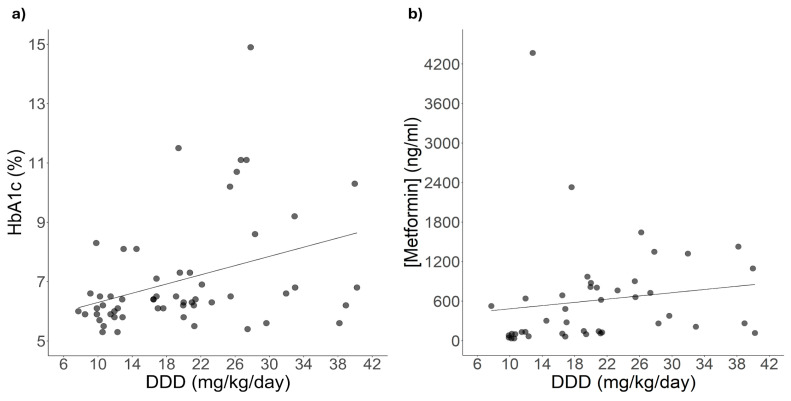
Effect of adjusted metformin DDD on HbA1c levels and metformin blood concentration. Correlation analysis of DDD and (**a**) HbA1c levels (n = 56, grey dots): HbA1c (%) = 5.5 + (0.077 × DDD), r^2^ = 0.128, *p* = 0.007 and (**b**) metformin blood concentrations (n = 43, grey dots): [Metformin] (ng/mL) = 361.240 + (12.230 × DDD, r^2^ = 0.019, *p* = 0.375).

**Figure 3 ijms-26-08652-f003:**
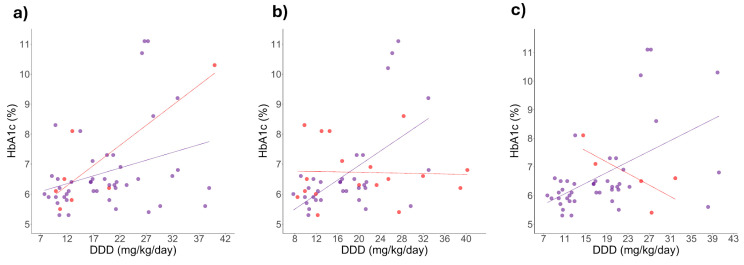
Effect of metformin adjusted dose on HbA1c levels according to a dominant genotypic model for *SLC47A1* and *SLC47A2*. Correlation analysis of HbA1c levels and metformin dose: (**a**) rs2289669: G homozygotes (n = 7, red dots): HbA1c (%) = 4.7 + (0.134 × DDD), r^2^ = 0.711, *p* value= 0.017 (red line) and A allele carriers (n = 45, purple dots): HbA1c (%) = 5.7 + (0.052 × DDD), r^2^ = 0.092 (purple line), *p* value = 0.042 ; (**b**) rs12945390: G homozygotes (n = 18, red dots): HbA1c (%) = 6.8 + (−0.003 × DDD), r^2^ = 0.001 (red line), *p* value = 0.887 and A allele carriers (n = 23, purple dots): HbA1c (%) = 4.5 + (0.121 × DDD), r^2^ = 0.330 (purple line), *p* value < 0.001; (**c**) rs34834489: A homozygotes (n = 5, red dots): HbA1c (%) = 9.1 + (−0.100 × DDD), r^2^ = 0.561 (red line), *p* value = 0.145 and Ga allele carriers (n = 43, purple dots): HbA1c (%) = 5.0 + (0.093 × DDD), r^2^ = 0.277 (purple line), *p* value < 0.001.

**Figure 4 ijms-26-08652-f004:**
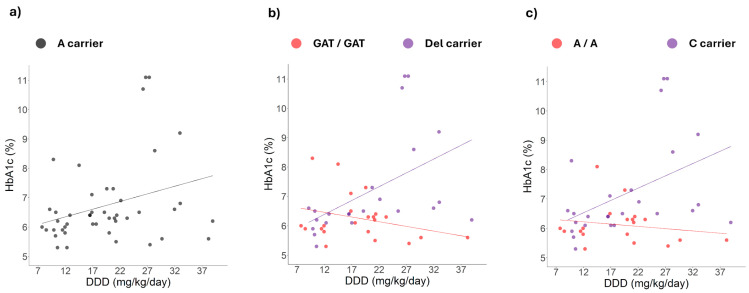
Effect of metformin dosage on HbA1c levels in dominant allele carriers of polymorphism rs2289669 in *SLC47A1* across different *SLC22A1* genotypes. Correlation analysis of HbA1c levels and metformin doses in (**a**) A allele carriers of rs2289669 (n = 45, grey dots): HbA1c (%) = 5.7 + (0.052 × DDD), r^2^ = 0.092, *p* value = 0.042, subdivided by polymorphism (**b**) rs72552763 [GAT homozygotes (n = 22, red dots): HbA1c (%) = 6.8 + (−0.031 × DDD), r^2^ = 0.087 (red line), *p* value = 0.184; del carriers (n = 23, purple dots): HbA1c (%) = 5.3 + (0.094 × DDD), r^2^ = 0.248 (purple line), *p* value = 0.016] or (**c**) rs622342 [A homozygotes (n = 19, red dots): HbA1c (%) = 6.4 + (−0.015 × DDD), r^2^ =0.031(red line), *p* value = 0.470; C allele carriers (n = 26, purple dots): HbA1c (%) = 5.5 + (0.002 × DDD), r^2^ = 0.083 (purple line), *p* value= 0.021].

**Figure 5 ijms-26-08652-f005:**
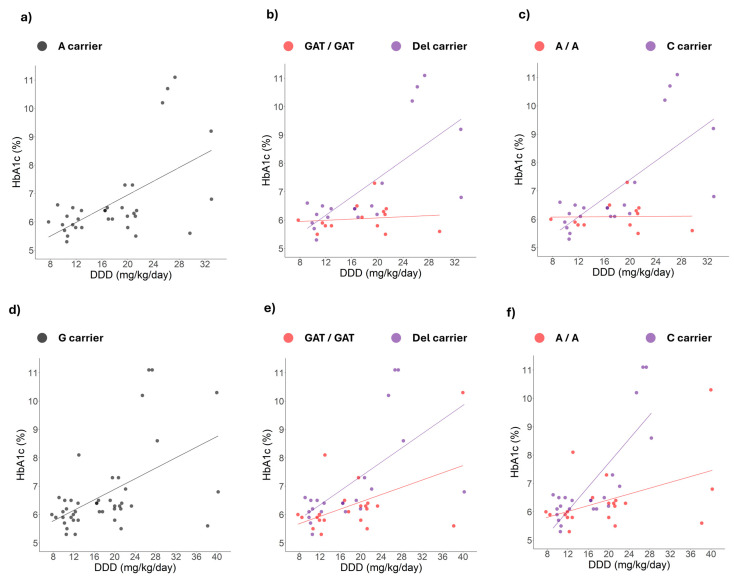
Effect of metformin dosage on HbA1c levels in dominant allele carriers of polymorphisms rs12943590 and rs34834489 in *SLC47A2* across different *SLC22A1* genotypes. Correlation analysis of HbA1C levels and metformin dose in (**a**) A allele carriers of rs12943590 (n = 33, grey dots): HbA1c (%) = 4.5 + (0.121 × DDD), r^2^ = 0.330, *p* value < 0.001, subdivided by polymorphism (**b**) rs72552763 [GAT homozygotes (n = 14, red dots): HbA1c (%) = 5.9 + (0.010 × DDD), r^2^ = 0.016 (red line), *p* value = 0.667; del carriers (n = 19, purple dots): HbA1c (%) = 4.2 + (0.161 × DDD), r^2^ = 0.510 (purple line), *p* value < 0.001] or (**c**) rs622342 [A homozygotes (n = 12, red dots): HbA1c (%) = 6.1 + (0.001 × DDD), r^2^ < 0.001 (red line), *p* value = 0.952; C allele carriers (n = 21, purple dots): HbA1c (%) = 4.1 + (0.164 × DDD), r^2^ = 0.520 (purple line), *p* value < 0.001]. Correlation analysis of HbA1C levels and metformin dose in (**d**) G allele carriers of rs34834489 (n = 43, grey dots): HbA1c (%) = 5.0 + (0.093 × DDD), r^2^ = 0.277, *p* value < 0.001, subdivided by polymorphism (**e**) rs72552763 [GAT homozygotes (n = 21, red dots): HbA1c (%) = 5.2 + (0.064 × DDD), r^2^ = 0.239 (red line), *p* value = 0.024; del carriers (n = 22, purple dots): HbA1c (%) = 4.8 + (0.115 × DDD), r^2^ = 0.379 (purple line), *p* value = 0.002] or (**f**) rs622342 [A homozygotes (n = 20, red dots): HbA1c (%) = 5.3 + (0.051 × DDD), r^2^ = 0.201 (red line), *p* value = 0.047; C allele carriers (n = 23, purple dots): HbA1c (%) = 3.5 + (0.211 × DDD), r^2^ = 0.651 (purple line), *p* value < 0.001].

**Table 1 ijms-26-08652-t001:** Allelic and genotypic frequencies of polymorphisms rs2252281, rs2289669, rs12943590, and rs34834489 in HRAEI DT2MM patients.

Gene	Polymorphisms	Allele	n	Frequency	Genotype	n	Frequency	^1^ *p* Value for HWE
*SLC47A1*	rs2252281 n = 204	T	363	0.890	TT	163	0.799	0.277
		C	45	0.110	TC	37	0.181	
					CC	4	0.020	
	rs2289669 n = 204	G	180	0.441	GG	43	0.211	0.349
		A	228	0.559	AG	94	0.460	
					AA	67	0.328	
*SLC47A2*	rs12943590 n = 204	G	175	0.429	GG	67	0.329	0.893
		A	233	0.57	AG	99	0.485	
					AA	38	0.186	
	rs34834489 n = 203	G	283	0.697	GG	101	0.498	0.431
		A	123	0.303	AG	81	0.399	
					AA	21	0.103	

^1^ *p* value corresponds to Pearson’s chi-square test for Hardy–Weinberg equilibrium.

## Data Availability

The data presented in this study are available on request via the corresponding author. These data are not publicly available because the patients and researchers are bound to an agreement establishing that only the head of the study and Mexican health authorities shall have access to them, in accordance with the presidential decree of 16 April 2015, sanctioning the General Law on Transparency and Access to Public Information.

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
