# Peer review of "Frequency of Polymorphisms in SLC47A1 (rs2252281 and rs2289669) and SLC47A2 (rs34834489 and rs12943590) and the Influence of SLC22A1 (rs72552763 and rs622342) on HbA1c Levels in Mexican-Mestizo Patients with DMT2 Treated with Metformin Monotherapy"

_ijms, 2025, doi:10.3390/ijms26178652_

Round 1
Reviewer 1 Report
Comments and Suggestions for Authors
This paper investigated the association of genotypes SLC47A1 (rs2252281 and rs2289669) and SLC47A2 (rs12943590 and rs34834489) with the therapeutic response to metformin when they are substratified according to different genotypes of rs72552763 and rs622342 in SLC22A1 of OCT1. This study is interesting, and the manuscript is interesting, however, I have some serious concerns about the present manuscript.
- Line 114-115. “…differ from those reported around the world” and Line 117 “different from those observed in the African population. Would the authors clarify if “around the world” not include “African population”?
- Line 112 – 117 and 121-127. There were redundancy in the text, making it hard to follow the argument; “differences respect to the world’s population in general” is vague; would the authors specify which statistical test was used and which differences were significant.
- Figure 1. The author stated that statistical differences relative to the DT2MM sample were assessed using a Chi-square test. In the figure, the diamond symbols appear to indicate differences among specific groups but not relative to the DT2MM sample. Would the authors clarify this?
- Line 152. The phase “except for size” was unclear and needed clarification.
- Figure S1. The sample sizes of some genotype groups were very small, making the analysis unreliable. In addition, the data appears Applying linear regression in the analysis is not justified. Did the authors consider alternative methods of analysis?
- Figure 2, the correlation analyses raise similar concerns. The regression lines and the results appears overstated.
- Figure 3 is misleading. The data was clustered and linear regression is not appropriate.
- Figure 4. Regression analysis on such small groups is statistically fragile and highly sensitive to outliers.
- Discussion 3.3. Lines 333-334. The authors stated that “…the significant and positive correlation between metformin dose and HbA1c levels suggested a treatment response failure in spite of increased doses”. This statement must be supported by clinical manifestation and justification of the reasons for treatment response failure.
- Discussion 3.4. Line 388. The statement “Our results evidence that not all polymorphisms may condition responses induced by other polymorphisms” is vague and not meaningfully supported.
- Limitations of the study should be discussed.
The authors use generally correct grammar and technical vocabulary, however, some sentences are long and redundant.
Author Response
RESPONSES TO REVIEWER´S OBSERVATIONS
Reviewer 1
This paper investigated the association of genotypes SLC47A1 (rs2252281 and rs2289669) and SLC47A2 (rs12943590 and rs34834489) with the therapeutic response to metformin when they are substratified according to different genotypes of rs72552763 and rs622342 in SLC22A1 of OCT1. This study is interesting, and the manuscript is interesting, however, I have some serious concerns about the present manuscript.
- Line 114-115. “…differ from those reported around the world” and Line 117 “different from those observed in the African population. Would the authors clarify if “around the world” not include “African population”?
R1: We have eliminated lines 112-117 to avoid confusion.
- Line 112 – 117 and 121-127. There were redundancy in the text, making it hard to follow the argument; “differences respect to the world’s population in general” is vague; would the authors specify which statistical test was used and which differences were significant.
R2: We appreciate your comment and agree with the pointed out redundancy; therefore, we have removed lines 112–117. In the footnote to Figure 1 (line 135 ), we indicated that the statistical test used for the comparisons between the observed genotypic frequencies in the HRAEI DT2MM patients and those reported in the 1000 Genomes Project for other populations was the Chi-square test. We also made modifications to the text to highlight the significant differences (lines 127, 129, 140).
- Figure 1. The author stated that statistical differences relative to the DT2MM sample were assessed using a Chi-square test. In the figure, the diamond symbols appear to indicate differences among specific groups but not relative to the DT2MM sample. Would the authors clarify this?
R3: We have modified the symbols used in Figure 1. The braces previously shown in Figures 1a–1c have been removed, and the single and double diamond symbols have been replaced with a single asterisk (*, p<0.05) and a double asterisk (**, p<0.001), indicating populations with genotypic frequencies that differ significantly from those observed in HRAEI DT2MM patients (135).
- Line 152. The phase “except for size” was unclear and needed clarification.
R4: The phrase “except for size” has been replaced with “except for height (m)” to avoid ambiguity. Height is the variable which, according to a Kruskal–Wallis test in Table 2s, was statistically different in at least one of the genotypes of the rs34834489 polymorphism (line 154).
- Figure S1. The sample sizes of some genotype groups were very small, making the analysis unreliable. In addition, the data appears Applying linear regression in the analysis is not justified. Did the authors consider alternative methods of analysis?
R5: Regression analyses aim to investigate the relationship between numerical variables [Bland, M. An Introduction to Medical Statistics; Oxford University Press, 2015; Petrie, A.; Sabin, C. Medical Statistics at a Glance; 4th ed.; John Wiley & Sons: Hoboken, NJ, 2020], such as HbA1c level, plasmatic metformin concentration, and daily metformin dose. Therefore, the use of a linear regression analysis is justified. Some authors suggest that at least 10 observations are required for each explanatory variable [Riley, R.D.; Ensor, J.; Snell, K.I.E.; Harrell, F.E.; Martin, G.P.; Reitsma, J.B.; Moons, K.G.M.; Collins, G.; van Smeden, M. Calculating the Sample Size Required for Developing a Clinical Prediction Model. BMJ 2020, m441, doi:10.1136/bmj.m441. Althubaiti, A. Sample Size Determination: A Practical Guide for Health Researchers. Journal of General and Family Medicine 2023, 24, 72–78, doi:10.1002/jgf2.600], or even fewer [Jenkins, D.G.; Quintana-Ascencio, P.F. A Solution to Minimum Sample Size for Regressions. PLOS ONE 2020, 15, e0229345, doi:10.1371/journal.pone.0229345]; this condition is met for eight out of the eleven regressions shown in Figure S1. Fewer than ten observations, as noted in your comment, may compromise the reliability of the model; however, this becomes problematic particularly when the aim is to predict the dependent variable from the linear regression model, as variance tends to be overestimated. In our case, we employed the linear regression model for exploratory purposes, with the aim of investigating the relationship between numerical variables. Nonetheless, to avoid a low number of observations and on biological grounds, we opted to explore subsequent analyses using a dominant genotypic model. Within this context, and in response to your valuable comment, we have added to the discussion section under the study limitations that findings from correlation analyses with fewer than ten observations should be interpreted with caution (lines 393-410). For these reasons, we do not consider the implementation of an alternative method.
6.- Figure 2, the correlation analyses raise similar concerns. The regression lines and the results appears overstated.
R6: We appreciate your concern regarding the potential overestimation of the results. The literature notes that two sources of overfitting in correlation models are: (1) the use of too many explanatory variables, and (2) an inadequate sample size for model construction [Aliferis, C.; Simon, G. Overfitting, Underfitting and General Model Overconfidence and Under-Performance Pitfalls and Best Practices in Machine Learning and AI. In; 2024; pp. 477–524. Babyak, M.A. What You See May Not Be What You Get: A Brief, Nontechnical Introduction to Overfitting in Regression-Type Models. Psychosomatic Medicine 2004, 66, 411–421, doi:10.1097/01.psy.0000127692.23278.a9]. The correlation analyses in Figure 2 are univariate, and therefore the first source of overfitting does not apply. As for the correlation models of DDD with HbA1c levels and plasmatic metformin concentration, these were respectively built upon 56 and 43 observations, thus complying with the recommendation of including more than 10 observations per explanatory variable [Riley, R.D.; Ensor, J.; Snell, K.I.E.; Harrell, F.E.; Martin, G.P.; Reitsma, J.B.; Moons, K.G.M.; Collins, G.; van Smeden, M. Calculating the Sample Size Required for Developing a Clinical Prediction Model. BMJ 2020, m441, doi:10.1136/bmj.m441. Althubaiti, A. Sample Size Determination: A Practical Guide for Health Researchers. Journal of General and Family Medicine 2023, 24, 72–78, doi:10.1002/jgf2.600]. Therefore, the overfitting risk in Figure 2 models is low, and so is consequently the likelihood of result overestimation. In this regard, the coefficient of determination obtained for the correlation (r² = 0.128) between DDD and HbA1c levels, which indicates a weak correlation, supports that results are not overestimated and was precisely what encouraged us to perform the subsequent analyses on genotype subgroups for the polymorphisms.
7.- Figure 3 is misleading. The data was clustered and linear regression is not appropriate.
R7: The variables HbA1c and DDD are numerical, which makes a linear regression analysis feasible, as one of its objectives is to evaluate the possible relationship between numerical variables, not solely to predict the dependent variable from the constructed model [Bland, M. An Introduction to Medical Statistics; Oxford University Press, 2015; Petrie, A.; Sabin, C. Medical Statistics at a Glance; 4th ed.; John Wiley & Sons: Hoboken, NJ, 2020]. The assumptions for performing a linear regression do not restrict its use on groups derived from a sample [3]. In Figure 3, we grouped the observations according to their genotype and obtained the linear regression individually for each group. We understand the potential for confusion, but in the figure legend, the correlation equation for each grouping is clearly presented, along with its coefficient of determination and the p-value corresponding to the linear regression analysis.
8.- Figure 4. Regression analysis on such small groups is statistically fragile and highly sensitive to outliers.
R8: We appreciate the reviewer’s observation and agree with it. However, as mentioned in the response to comment five, using at least ten observations per explanatory variable is considered adequate for conducting a correlation analysis. The increase in the number of observations was achieved by applying the dominant genotypic model. It is important to note that employing very large sample sizes is not considered a recommended practice, as this can result in p-values below the significance threshold, even when the effect is not clinically relevant [Althubaiti, A. Sample Size Determination: A Practical Guide for Health Researchers. Journal of General and Family Medicine 2023, 24, 72–78, doi:10.1002/jgf2.600. Faizi, N.; Alvi, Y. Biostatistics Manual for Health Research: A Practical Guide to Data Analysis; Elsevier, 2023].
9.- Discussion 3.3. Lines 333-334. The authors stated that “…the significant and positive correlation between metformin dose and HbA1c levels suggested a treatment response failure in spite of increased doses”. This statement must be supported by clinical manifestation and justification of the reasons for treatment response failure.
R9: We have restructured lines 333 and 334 as follows: the significant positive correlation between DDD and HbA1c levels suggests that, in some patients, an increase in the metformin dose does not correlate with a decrease in HbA1c. (lines 329-331).
10.- Discussion 3.4. Line 388. The statement “Our results evidence that not all polymorphisms may condition responses induced by other polymorphisms” is vague and not meaningfully supported.
R10: We have modified line 388 as follows: our results evidence that not all of the polymorphisms we studied, either individually or in combination, exhibit any type of correlation between metformin dose and HbA1c. (line 386-388).
11.- Limitations of the study should be discussed
R11: We have added a discussion of the study limitations to the manuscript (lines 393-410).
Our study presents several limitations that should be considered both when interpreting its findings and also to guide the design of future studies. The sample size (n = 204) was determined based on the reported global proportions of the polymorphisms, addressing the primary objective of determining the genotypic frequencies of polymorphisms rs2289669, rs2252281, rs12943590, and rs34834489, which was achieved. When stratifying the genotypes of the studied polymorphisms, some groups comprised fewer than 10 observations (GG of rs2289669 and AA of rs34834489); therefore, the correlation and regression analyses between HbA1c and DDD should be interpreted with caution. On the other hand, the sample size for correlations of the remaining genotypes met methodological recommendations [Riley, R.D.; Ensor, J.; Snell, K.I.E.; Harrell, F.E.; Martin, G.P.; Reitsma, J.B.; Moons, K.G.M.; Collins, G.; van Smeden, M. Calculating the Sample Size Required for Developing a Clinical Prediction Model. BMJ 2020, m441, doi:10.1136/bmj.m441. Althubaiti, A. Sample Size Determination: A Practical Guide for Health Researchers. Journal of General and Family Medicine 2023, 24, 72–78, doi:10.1002/jgf2.600]. Nevertheless, it was insufficient for performing multivariate analyses that could provide additional information.
Another limitation is that our study has a cross-sectional design, and we only have a single time-point measurement of the biomarkers. Consequently, it is not possible to determine whether the findings remain consistent over time. Finally, the interval between the collection of peripheral venous blood samples and the evening metformin dose could vary between patients, potentially leading to wide variability in plasma metformin concentration.

Reviewer 2 Report
Comments and Suggestions for Authors
The study addresses a relevant pharmacogenetic question in an understudied population. Below are my comments:
1- The study focuses exclusively on metformin both in its pharmacogenetic hypothesis and in the cohort restricted to individuals receiving metformin monotherapy, yet the current title is vague and does not clearly reflect this focus.
2- While the introduction gives a strong background on metformin transporters and genetic polymorphisms, it does not clearly explain why only these four specific SNPs were chosen as the main focus as other variants are also well-known to influence metformin response.
3- The study is cross-sectional, yet the manuscript sometimes implies causal relationships. This design cannot establish causality, only association. The language should be revised accordingly.
4- Blood sampling "within an interval of 8 h after the evening’s metformin dose" is too vague. An 8-hour window is quite wide in pharmacokinetic terms.
5- Many genotype subgroups had very low sample size. The statistical power to detect true associations or reliably estimate correlation coefficients is therefore very low. Moreover, the study performs many statistical tests, but no multiple-testing correction is applied.
6- A positive correlation between daily dose of metformin and HbA1c is interpreted as “treatment response failure.” While plausible, such a correlation could just reflect clinical titration behavior (patients with higher HbA1c will receive higher doses), it is not mechanistically telling without longitudinal change data.
7- The discussion does not contain acknowledgment of limitations.
8- The gene-gene interaction interpretations (Lines 381-382) are highly speculative and lack direct mechanistic support.
Author Response
Reviewer 2
The study addresses a relevant pharmacogenetic question in an understudied population. Below are my comments:
- The study focuses exclusively on metformin both in its pharmacogenetic hypothesis and in the cohort restricted to individuals receiving metformin monotherapy, yet the current title is vague and does not clearly reflect this focus.
R1: We have modified the title aiming towards a clearer and more direct connection to the article (lines 2-6).
Frequency of polymorphisms in SLC47A1 (rs2252281, rs2289669) and SLC47A2 (rs34834489, rs1208969), and the influence of SLC22A1 (rs72552763, rs622342) on HbA1c levels in Mexican-mestizo patients with DMT2 treated with metformin monotherapy.
While the introduction gives a strong background on metformin transporters and genetic polymorphisms, it does not clearly explain why only these four specific SNPs were chosen as the main focus as other variants are also well-known to influence metformin response.
R2: We understand your concern regarding the restriction of the study to the analysis of the four selected variants; therefore, we have added to the manuscript text the rationale that led us to study these polymorphisms (lines 86-92.)
- The study is cross-sectional, yet the manuscript sometimes implies causal relationships. This design cannot establish causality, only association. The language should be revised accordingly.
R3: Thank you for your comment; we fully agree and have therefore carefully reviewed the manuscript and revised the sentences that could potentially cause the confusion you noted (lines 355-356, 369-370, 560-564).
MODIFIED TEXT 1:
Finally, the positive DDD-HbA1c correlations observed through a dominant genotypic model in both G homozygotes and A of rs2289669, differ from previous studies where A was reported as the protection allele [35,36]. When isolated, none of our results found rs2289669 alleles correlating DDD and HbA1c levels; however, as evidenced by the simultaneous polymorphism analysis, the observations from previous reports could be due to the presence of several polymorphisms across different transporter genes at the same time.
MODIFIED TEXT 2:
This suggests a possible influence by these polymorphisms on HbA1c levels. Our results match the increased reduction in HbA1c levels reported by rs2289669 allelic variant carriers (SLC47A1) who are also rs622342 allelic variant homozygotes (SLC22A1) [23].
MODIFIED TEXT 3:
The individual analysis of rs2252281, rs2289669, rs12943590, and rs34834489 did not yield an association[j1] with therapeutic response. However, the collective analysis including the polymorphic variants of SLC22A1, rs72552763, and rs622342 approached through a dominant genotypic model allowed us to identify allelic combinations which correlate DDD and HbA1c.
…Thereby, this particular allele combination might be relevant for therapeutic response as estimated by HbA1c levels The rs34834489 reference allele in SLC47A2 could imply a non-control risk, given that the correlation between HbA1c levels and metformin daily dose remained significant and positive when its carriers were also carrying any rs72552763 and rs62234 alleles inSLC22A1.
4- Blood sampling "within an interval of 8 h after the evening’s metformin dose" is too vague. An 8-hour window is quite wide in pharmacokinetic terms.
R4: We appreciate the reviewer’s comment and would like to make the following:
- We have added to the discussion section that the eight-hour interval may be wide and even vary between patients, as this is an observational study and we do not have control over this variable, as indicated in section 4.4 of the manuscript “Patients were summoned by their respective treating physician having observed a fasting period of at least 8 h”. (lines 393-410).
- The primary objective of our tudy was to determine the genotypic frequencies of the polymorphisms rs2252281, rs2289669, rs12943590, and rs34834489, as well as to explore their possible relationship with the therapeutic response to metformin monotherapy.
5. Many genotype subgroups had very low sample size. The statistical power to detect true associations or reliably estimate correlation coefficients is therefore very low. Moreover, the study performs many statistical tests, but no multiple-testing correction is applied.
R5: Some authors suggest that at least 10 observations are required per explanatory variable for a correlation analysis [2,3]. In our initial analysis, Figure 1s, eight of the eleven regressions met this criterion. Additionally, during our data analysis, to avoid a low number of observations and on biological grounds, we implemented a dominant genotypic model approach for the subsequent analyses, Figures 3, 4, and 5. Inevitably, the correlation analysis for the G homozygotes of rs2289669 and the A homozygotes of rs34834489 was conducted with fewer than 10 observations, which we have acknowledged in the discussion as one of the study’s limitations. Furthermore, correlation analyses aim to evaluate the possible relationship between numerical variables, so obtaining a low correlation coefficient indicates that the independent variable in the models does not largely explain the dependent variable; however, it is noteworthy that a correlation between the variables does exist. To attempt a better model fit, multivariate analyses could have been performed, but the number of available observations was a limitation. Consequently, this has also been added as one of the study’s limitations. Regarding multiple correlations, initially, given the absence of significant differences, correction for multiple comparisons was not considered. However, in response to your comment, Tables S3–S5 have been added to the supplementary material, including the multiple comparison correction performed using the Bonferroni method. It should be noted that, after applying the Bonferroni correction for multiple comparisons, no significant differences were found.
- A positive correlation between daily dose of metformin and HbA1c is interpreted as “treatment response failure.” While plausible, such a correlation could just reflect clinical titration behavior (patients with higher HbA1c will receive higher doses), it is not mechanistically telling without longitudinal change data.
R6: We appreciate the reviewer’s comment and, in response, we can note that due to the cross-sectional design of this exploratory study, we do not have data on HbA1c changes over time, as would be the case in a longitudinal study. In this regard, the discussion highlights that the cross-sectional nature of the study is one of its main limitations. We do not rule out conducting a longitudinal study in the future, with a larger sample size and taking into account the findings of the present study, in order to provide mechanistic evidence.
7. The discussion does not contain acknowledgment of limitations.
R7: We appreciate your observation and have added a discussion of the study’s limitations to the manuscript. (lines 393-410).
- The gene-gene interaction interpretations (Lines 381-382) are highly speculative and lack direct mechanistic support.
R8: We appreciate the reviewer’s comment and have revised the phrasing of lines 381–382 in the first version of the manuscript, which now reads as follows (lines 377-385):
The increment in SLC47A2 translation [15,21] paired with non-functional SLC22A1 alleles could lead to an augmented metformin elimination, in view of the effect combination. The lower hepatic volume of distribution observed in carriers of the DEL allele of rs72552763 [22] could increase the plasma concentration of metformin. When this is combined with an increase in metformin renal clearance, as observed in carriers of the variant allele of polymorphisms rs12943590 or rs34834489 [15,21], it could promote metformin elimination, thereby reducing its therapeutic efficacy, since metformin’s main action mechanism is the reduction of hepatic glucose production [23].
Round 2
Reviewer 1 Report
Comments and Suggestions for Authors
I have no further comments with the manuscript.
Reviewer 2 Report
Comments and Suggestions for Authors
The authors have adequately addressed the comments.